# Evaluation of the Impact of Imprinted Polymer Particles on Morphology and Motility of Breast Cancer Cells by Using Digital Holographic Cytometry

**Megha Patel [1], Marek Feith [2], Birgit Janicke [3], Kersti Alm [3] and Zahra El-Schich [1,*]**

[1] Department of Biomedical Sciences, Faculty of Health and Society, Malmö University, 205 06 Malmö, Sweden; drmeghabhavin@gmail.com

[2] Department of Physiology, Faculty of Medicine, Masaryk University, 625 00 Brno, Czech Republic; mafeith@seznam.cz

[3] Phase Holographic Imaging AB, 223 63 Lund, Sweden; birgit.janicke@phiab.se (B.J.); kersti.alm@phiab.se (K.A.)

[*] Correspondence: zahra.el-schich@mau.se

**Abstract:** Breast cancer is the second most common cancer type worldwide and breast cancer metastasis accounts for the majority of breast cancer-related deaths. Tumour cells produce increased levels of sialic acid (SA) that terminates the monosaccharide on glycan chains of the glycosylated proteins. SA can contribute to cellular recognition, cancer invasiveness and increase the metastatic potential of cancer cells. SA-templated molecularly imprinted polymers (MIPs) have been proposed as promising reporters for specific targeting of cancer cells when deployed in nanoparticle format. The sialic acid-molecularly imprinted polymers (SA-MIPs), which use SA for the generation of binding sites through which the nanoparticles can target and stain breast cancer cells, opens new strategies for efficient diagnostic tools. This study aims at monitoring the effects of SA-MIPs on morphology and motility of the epithelial type MCF-7 and the highly metastatic MDAMB231 breast cancer cell lines, using digital holographic cytometry (DHC). DHC is a label-free technique that is used in cell morphology studies of e.g., cell volume, area and thickness as well as in motility studies. Here, we show that MCF-7 cells move slower than MDAMB231 cells. We also show that SA-MIPs have an effect on cell morphology, motility and viability of both cell lines. In conclusion, by using DH microscopy, we could detect SA-MIPs impact on different breast cancer cells regarding morphology and motility.

**Keywords:** breast cancer; digital holographic cytometry; molecularly imprinted polymers; motility; sialic acid; viability

## 1. Introduction

Breast cancer is the second most widespread type of cancer globally and affects 12.5% of all women during their lifetime [1]. Breast cancer metastasis causes the majority of deaths from breast cancer. Survival of breast cancer highly depends on the presence of metastatic cells and tumour grade. Despite observable progress in diagnosis methods, early detection of cancer dissemination seems to be challenging, therefore new approaches for early detection of metastatic cells are required [2–4].

Cellular morphology plays a crucial role in cancer invasiveness and forming of metastases. During epithelial-to-mesenchymal transition (EMT), cells undergo many biochemical changes including cellular shape shift and changes in mass distribution to acquire a mesenchymal phenotype. These changes allow cells to migrate through epithelial tissue and become circulating and eventually metastatic tumour cells. These cells express various amounts of surface glycans in order to attach extrinsic tissue, disseminate and thereafter form metastases [5,6].

Glycosylation plays an important role in cell biology. Cell surface glycoproteins mediate cell-cell communication, cell adhesion and migration [7]. Their functions were described previously in inflammatory processes, which occur also in cancer, as they contribute to tumour progression, angiogenesis and metastasis formation [8]. Glycosylation is one of the most common post-translational protein modifications and may be present in a wide range of forms (fucosylation, truncation, branching and sialylation). Recent studies suggest that sialic acid (SA) can contribute to cellular recognition, cancer invasiveness and increase the metastatic potential of cancer cells [7,9]. SA is the terminating monosaccharide on glycan chains of glycosylated proteins. SA residues have been reported to be highly expressed on the surfaces of cancer cells, which make them promising candidates as targets and cancer-specific antigens [10]. The residues provide two distinct patterns that can be utilized for even more specific characterization of cancer cells as SA is linked through $\alpha$-2,3 or $\alpha$-2,6 [9].

Molecularly imprinted polymers (MIPs) have been proposed as promising reporters for specific targeting of cancer cells [11]. MIPs are synthetically prepared particles, with high binding specificity and affinity to the target molecule. They are synthetic alternatives to lectins and antibodies and can be endowed with secondary functions such as luminescence, magnetic or electrochemical properties in a straightforward manner. MIPs are also comparatively easy to prepare, commonly at lower costs than antibodies [12–14]. MIPs targeting SA could contribute to solving challenges faced in the tailor-making of macromolecular reporters, such as poor specificity because of the use of entire cancer cells as templates [15]. We and others have assessed the use of SA as template in the preparation of MIPs to detect SA on cancer cells [16–20].

Digital holographic cytometry (DHC) has been demonstrated as an effective tool for long-term label-free cell observations and evaluation of cell morphological and dynamical parameters in vitro [21,22]. DHC can be also used for distinction and determination of cellular processes such as cell death [23–25], cell cycle [25], cellular uptake [26], and cell morphology changes [22]. Mölder et al. reported that DHC is a non-invasive, label-free cell counting and quantitative technique for analysing adherent cells [27]. Kemper et al. reviewed new ways of monitoring the cellular morphology changes in response to drugs [28] and Yu et al. derived conclusions on whether cells were motile or not with results from quantitative cell analysis using DHC [21]. Cell morphology and motility can be studied by segmentation of the cell outlines, and differences in cell types can be detected because of the changes in the morphology or migration patterns in drug response [24,29–31]. The principle of DHC is based on the creation of interference patterns as light is transmitted through the cells. A laser beam is divided into a reference and a sample beam, and both beams are later merged again to create the raw image. The changes in interference pattern are assessed after capture on a CCD camera. Digital autofocus and reconstruction of the 3D-image of the sample is provided by computer algorithms [32,33].

In this study, we aim to evaluate the impact of sialic acid-molecularly imprinted polymers (SA-MIPs) on the motility and viability of two breast cancer cell lines MCF-7 and MDAMB231. Flow cytometry was used to evaluate the level of SA using MIPs and the lectins MALI (from Maackia amurensis) and SNA (from Sambucus nigra). The lectins MALI bind specifically to $\alpha$-2,3 SA and SNA bind specifically to $\alpha$-2,6 SA. DHC was used to quantify cell morphology and motility to reveal the metastatic potential of the cell lines. Viability assays were performed on the MCF-7 and MDAMB231 cells lines treated or not treated with SA-MIPs. Here, we show that MDAMB231 cells had higher motility compared to MCF-7 cells and that SA-MIPs treatment increased the motility and slightly decreased the viability for MCF-7 cells, while for MDAMB231 cells the motility decreased and the viability increased.

## 2. Materials and Methods

### 2.1. Cell Lines and Culture

Two different human breast cancer cell lines, MCF-7 and MDAMB231 were obtained from American Type Culture Collection (ATCC/LGC Standards, Teddington, UK). The MCF-7 cell line was

cultured in RPMI1640 medium supplemented with 10% foetal bovine serum (FBS) (Thermo Fisher Scientific, Waltham, MA, USA) and gentamycin, 50 μg/mL (Sigma-Aldrich, St Louis, MO, USA). The MDAMB231 cell line was cultured in DMEM (Thermo Fisher Scientific) supplemented with 10% FBS. The cell lines were cultured at 37 °C with 5% $CO_2$ in 95% humidity.

## 2.2. Flow Cytometry Analysis of SA-MIPs and Lectins

SA-MIPs with a diameter of 200 nm were synthesized and prepared as described previously [20]. Cells were harvested by trypsinization and $1 \times 10^6$ cells/sample were stained with 0.04 mg/mL of SA-MIPs in phosphate buffered saline (PBS, Thermo Fisher Scientific) or left unstained as a control with only PBS and incubated at 4 °C for 30 min. Thereafter the cells were washed twice with 2 mL of PBS and analysed using flow cytometry (BD Biosciences, Accuri C6 Flow Cytometry, NJ).

For staining with the biotin-conjugated lectins MALI (and SNA (cells were harvested and $5 \times 10^5$ cells/sample were stained with 5 μg/mL of MALI or SNA (Vector Laboratories, Burlingame, CA, USA) diluted in PBS or left unstained as a control with only PBS and incubated at 4 °C for 30 min. Thereafter, the cells were washed twice with 2 mL of PBS and incubated with streptavidin-FITC (Sigma), diluted in PBS to 1/100 at 4 °C for 20 min before washing twice with 2 mL of PBS and analysed with flow cytometry.

## 2.3. DHC and Computer Software

For each cell line, $2 \times 10^4$ cells/well were seeded in a working volume of 1.8 mL in Sarstedt's lumox® multiwell 24 wells plate (Sarstedt, Inc. Nümbrecht, Germany). The cells were incubated for 24 h at 37 °C in humidified 5% $CO_2$ atmosphere to allow to adhere. After incubation, 0.04 mg/mL of SA-MIPs were added to the wells. Control wells were left without additive for both cell lines. The plate was incubated for 1 h at 37 °C in humidified 5% $CO_2$ atmosphere. After the incubation, the standard lid was replaced with the HoloLid™ (Phase Holographic Imaging AB, Lund, Sweden). The plate was mounted on a HoloMonitor™ M4 (Phase Holographic Imaging, AB), which is placed inside a cell incubator to ensure stable conditions of the cells during long-term experiments. The HoloMonitor™ is equipped with a 635 nm low laser beam with 0.2 mW/cm² to prevent phototoxicity effects on the cells. The image analysis was performed with the proprietary AppSuite software (Phase Holographic Imaging AB). The images were acquired automatically every 30 min for 72 h.

## 2.4. Cell Viability Assay

A total of $5 \times 10^3$ cells of MCF-7 and MDAMB231 cell lines were seeded in parallel 96-well plates with 100 μL of DMEM or RPMI medium respectively and incubated for 24 h to allow the cells to adhere at 37 °C with 5% $CO_2$ in 95% humidity. The cells were then incubated with 0.04 mg/mL of SA-MIPs or medium as control and further incubated for 24 h, 48 h, and 72 h before analysis. MTS-assay was performed by adding 20 μL of MTS/PMS solution (CellTiter 96® AQueous Non-Radioactive Cell Viability Assay, Assay, Promega Corporation, Madison, WI, USA) to the wells and the plates were incubated for two more hours. The absorbance was measured at 490 nm with BIO-TEK® micro plate reader.

## 2.5. Statistical Analysis

Mean and relative standard deviations, expressed as the coefficient of variation (CV), were used for the statistical analysis for all calculations. For the cell motility studies, at least five DHC images per sample in different positions were captured for the experiment. The cell morphology analysis was measured through the mean and CV of the cell mean volume, mean area and mean thickness of the replicates. Experiments were repeated on separate days, at least three times. Data for DHC and MTS-assay are expressed as mean ± standard deviation (STDEV). Statistical significance was determined by two-tailed unpaired Student's *t*-test with *p* values ≤ 0.05 considered to be significant.

## 3. Results

### 3.1. Different Expression Patterns for SA

The mean fluorescence intensity (MFI) of the MCF-7 cells and MDAMB231 cells after staining with 0.04 mg/mL of SA-MIPs is presented in histograms. The MCF-7 cells express higher SA levels as detected with SA-MIPs compared to MDAMB231. The MCF-7 cells express low $\alpha$-2,6 SA levels and high $\alpha$-2,3 SA levels, while MDAMB231 express high $\alpha$-2,6 SA and $\alpha$-2,3 SA levels as detected with MALI and SNA (Figure 1).

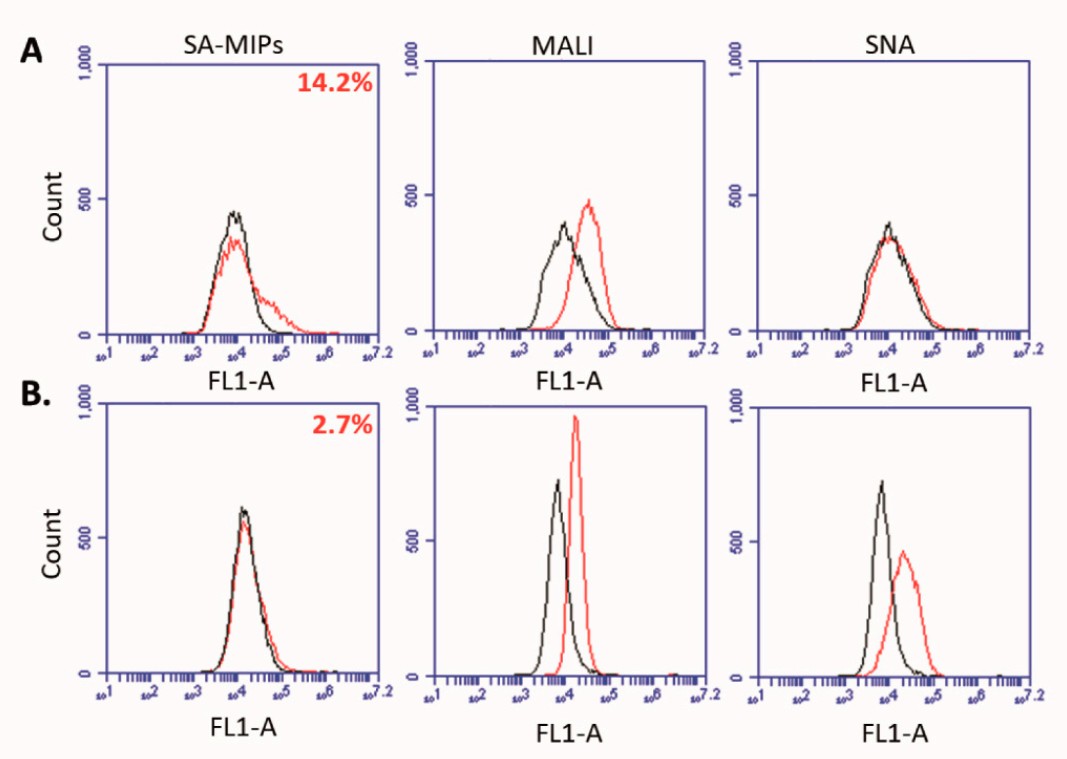

**Figure 1.** Histograms presenting the mean fluorescence intensity (MFI) for cell staining with sialic acid-molecularly imprinted polymers (SA-MIPs) or the lectins MALI ($\alpha$-2,3-specific) and SNA ($\alpha$-2,6-specific) for both MCF-7 cells (**A**) and MDAMB231 cells (**B**). The black traces show the unstained samples, while the red traces show the stained samples. The experiment is repeated at least three times with similar results. The histograms show results from one representative experiment.

### 3.2. Different Motility between the Cell Lines Analysed with DHC

The motility analysis with DHC showed differences between MCF-7 and MDAMB231. The MDAMB231 cells had higher motility (Video S3) compared to the MCF-7 cells (Video S1) with up to twice the accumulated distance. The MDAMB231 cells incubated with SA-MIPs showed a slight decrease of the accumulated distance after 72 h, while for the MCF-7 cells the accumulated distance increased (Figure 2, Videos S2–S4).

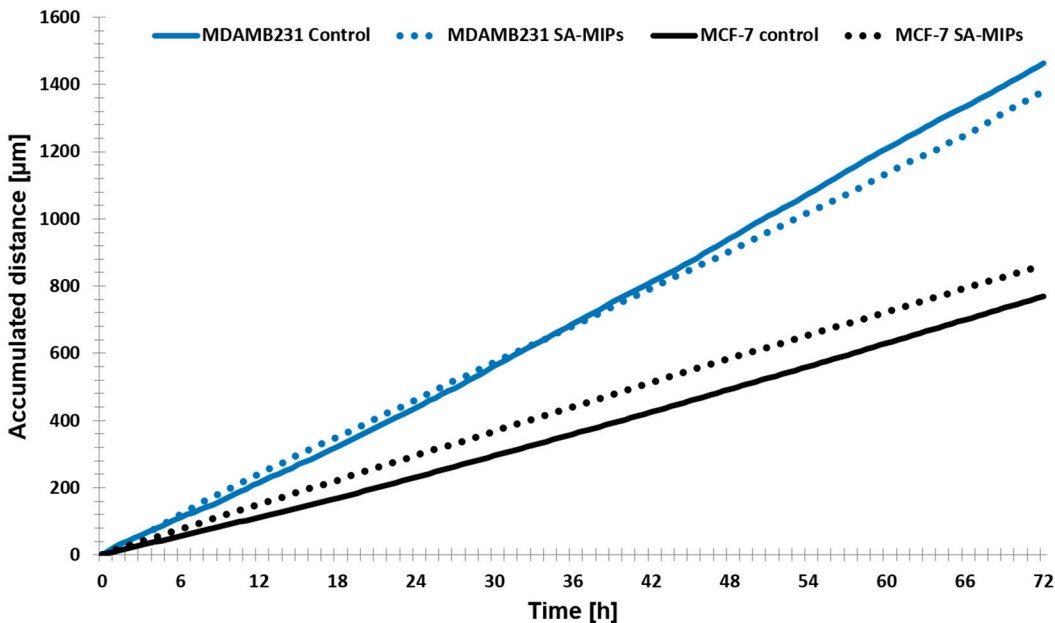

**Figure 2.** Motility diagram presenting the accumulated distance in μm for the untreated and treated MCF-7 cells and MDAMB231 cells with 0.04 mg/mL of SA-MIPs incubated over time. The holograms were captured every 30 min for 72 h. The experiment is repeated at least three times with similar results. The diagram show results from one representative experiment.

### 3.3. Morphology Changes Detected with DHC

MCF-7 and MDAMB231 cells (Figure 3B,D) treated with SA-MIPs for 72 hours had increased in thickness compared to untreated MCF-7 and MDAMB231 cells, respectively as detected by DHC (Figure 3A,C).

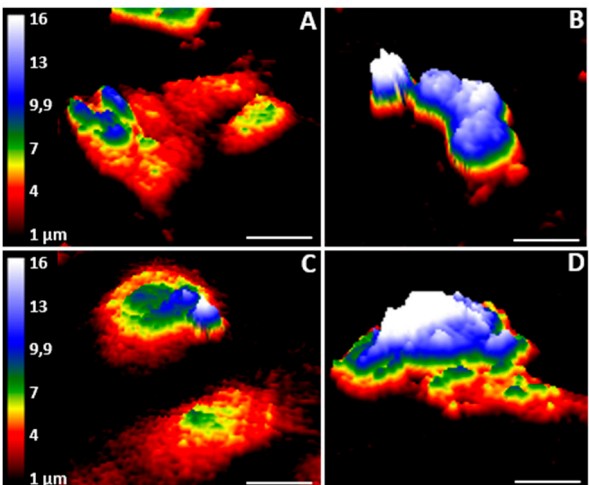

**Figure 3.** 3D holograms presenting the untreated MCF-7 cells (**A**), MDAMB231 (**C**) and the treated MCF-7 cells (**B**), MDAMB231 (**D**) with 0.04 mg/mL of SA-MIPs after 72 h incubation. The colour scale represents the thickness and the scale bar represents 20 μm.

Both treated MCF-7 cells and MDAMB231 cells increased in mean volume compared to untreated cells (Table 1). The morphology changes differ between the two cell lines. For treated MCF-7 cells, the mean area decreased, and the mean thickness increased compared to the untreated cells. For treated MDAMB231 cells, the mean area increased and the mean thickness slightly decreased compared to the untreated cells (Table 1).

**Table 1.** Morphology parameters detected with DHC for MCF-7 cells and MDAMB231 cells after 72 h of SA-MIPs treatment. The results represent mean value of five positions in the wells for each condition. Statistical significance was determined with *p* values ≤ 0.05.

| | Mean Volume [μm$^3$] (±cv%) | *p* value | Mean Area [μm$^2$] (±cv%) | *p* Value | Mean Thickness [μm] (±cv%) | *p* Value |
|---|---|---|---|---|---|---|
| MCF-7 Control | 2600 (±15) | **0.027** | 685 (±15) | 0.306 | 3.8 (±18) | **0.024** |
| MCF-7 SA-MIPs | 3190 (±11) | | 628 (±8) | | 4.8 (±10) | |
| MDAMB231 Control | 2825 (±4) | **0.005** | 574 (±21) | 0.247 | 5.1 (±6) | 0.797 |
| MDAMB231 SA-MIPs | 3471 (±15) | | 666 (±11) | | 5.0 (±14) | |

## 3.4. Cell Viability Variation Over Time for SA-MIPs Treated Cell Lines

MTS assay was performed to investigate the viability pattern and the effect of the SA-MIPs on cell viability over time. MCF-7 and MDAMB231 cells were treated with 0.04 mg/mL SA-MIPs or left untreated as control and analysed after 24, 48 and 72 h. The cell viability had decreased significantly for treated MCF-7 cells compared to untreated cells after 24, 48 and 72 h. For the MDAMB231 cells, cell viability had significantly increased for treated cells compared to untreated cells after 72 h (Figure 4).

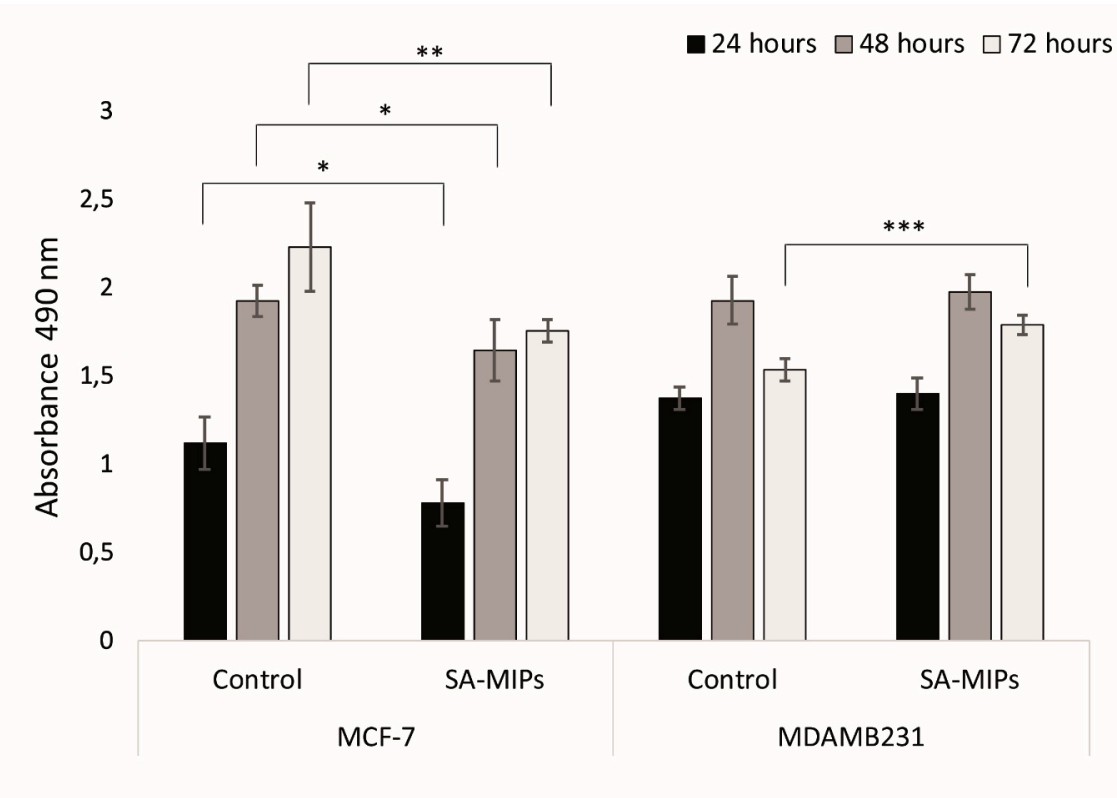

**Figure 4.** Cell viability assay with MCF-7 cells and MDAMB231 cells treated with 0.04 mg/mL of SA-MIPs or left as a control for three time points, 24, 48 and 72 h. The absorbance was measured at 490 nm. The results represent the mean value of five replicate wells for each condition and repeated at least twice with similar results. Statistical significance was determined with *p* values ≤ 0.05 *, ≤0.01 **, ≤0.001 ***.

## 4. Discussion

Breast cancer is a heterogeneous disease consisting of different subtypes with varying prognosis [34]. Breast cancer treatment resistance is common and increases mortality [35]. Tumour cells produce increased levels of SA [7]. It has been reported that overexpression of SA in metastatic cancer controls the tumour cell growth and differentiation by interfering with neural cell adhesion molecule signalling

at cell-cell contacts [36]. The MCF-7 cell line expresses high SA levels, mainly with α-2,3 linkage, and was detected to a higher extent with SA-MIPs and MALI using flow cytometry, while MDAMB231 expresses high SA (α-2,3) levels as detected with MALI and SNA.

In this study, we investigated the cell motility using DHC for the two breast cancer cell lines MCF-7 and MDAMB231. Here, we show that MDAMB231 cells have higher motility compared to MCF-7 cells. These observations are in agreement with the existing evidence that show that MDAMB231 cells are more motile and invasive than the MCF-7 cells, which are generally less invasive and have poor migration capabilities [37]. The MCF-7 cell line represents a non-motile epithelial cell line with the ability to retain the ideal features of the mammary epithelium with low-risk of metastasis [37,38]. The MDAMB231 cell line is a highly motile, aggressive, invasive and poorly differentiated triple-negative breast cancer cell line. The MDAMB231 cell line mediates proteolytic degradation in the extracellular matrix and contributes to invasiveness and metastasis formation by varying morphological characteristics [39,40].

The two cell lines showed different morphologies and different motility patterns. The cell volume significantly increased after SA-MIPs treatment for both cell lines, suggesting ingestion or binding of SA-MIPs. These results agree well with a recent study in which we showed that the increase in cell size of macrophages was a response to SA-MIPs treatment [41]. Peter et al. have shown that the very thin outer part of the cells are not detected by DH microscopy, and therefore the cell volume measurements can be misleading [42]. The outer parts of the cells used in the present study can be detected using the HoloMonitor as the cells are thicker and do not spread as much as the cells used by Peter et al.

The findings of this study showed that SA-MIPs affected the cell viability of the cell lines at different stages during the experiment. Here, a decrease in cell viability for the MCF-7 cell line treated with SA-MIPs after 24 h, 48 h and 72 h, correlated inversely with increased cell motility. There was an increase in the cell viability of the MDAMB231 cell line treated with SA-MIPs at 72 h, which correlated inversely with decreased cell motility. The increase in the cell viability of MDAMB231 cells could be due to the high expression of α-2,6 and α-2,3 SA, which could lead to increased proliferation of the breast cancer cells [43]. The increase in viability analysed with MTS might be explained by the increase in cell size, as larger cells acquire higher metabolism. The MTS method, being an indirect analysis method of viability, is based on cell metabolism that in some cases give false results because of changes in metabolism rather than cell numbers.

## 5. Conclusions

SA-MIPs affect motility, morphology and viability on both MCF-7 and MDAMB231 cell lines. The MDAMB231 cells showed higher motility compared to the MCF-7 cells but with SA-MIPs treatment the motility decreased for the MDA-MB231 and increased for the MCF-7 cells. The SA-MIPs treatment over time affected cell morphology such as cell volume, area and thickness. In conclusion, DHC is a powerful tool to analyse the motility and morphology differences between cell lines as well as the SA-MIPs-induced cell response.

**Supplementary Materials:** The following are available online at http://www.mdpi.com/2076-3417/10/3/750/s1. Video S1: MCF-7-untreated cells. Video S2: MCF-7 SA-MIPs-treated cells. Video S3: MDAMB231-untreated cells. Video S4: MDAMB231 SA-MIPs-treated cells.

**Author Contributions:** Methodology, M.P. and M.F.; software, M.P. and Z.E.-S.; data curation, M.P.; B.J. and Z.E.-S.; writing—original draft preparation, M.P., M.F. and Z.E.-S.; writing—review and editing, K.A. and B.J.; supervision, Z.E.-S. All authors have read and agreed to the published version of the manuscript.

**Funding:** This research was funded by grants from The Swedish Knowledge Foundation, Malmö University, and Biofilms—Research Center for Biointerfaces at Malmö University, Sweden.

**Acknowledgments:** We thank M. W. Kimani and K. Rurack (BAM, Berlin) for providing the SA-MIPs.

**Conflicts of Interest:** The authors declare no conflict of interest.

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
