# Peer review of "Evaluation of the Impact of Imprinted Polymer Particles on Morphology and Motility of Breast Cancer Cells by Using Digital Holographic Cytometry"

_applsci, doi:10.3390/app10030750_

Round 1

Reviewer 1 Report

The present ms describes the investigation of cancer cell morphology and motility after nanoparticle exposure. The authors use a powerful method, digital holographic microscopy to follow cellular changes in real time in a completely label-free manner. The topic is worth investigating and the overall results are sound.

I suggest publication after the below points addressed in full.

I think the end of the abstract, explaining the results, are too general. ( "have an impact", "affects different cells ... differently"). I suggest to try to be more specific here. Some of the figure letters are too small to read well. When explaining DHC the following paper should be cited, especially because it uses exactly the same device, and it explains well the limitations of the technique, and employs it in interesting new directions. https://www.spiedigitallibrary.org/journals/Journal-of-Biomedical-Optics/volume-20/issue-06/067002/Incubator-proof-miniaturized-Holomonitor-to-in-situ-monitor-cancer-cells/10.1117/1.JBO.20.6.067002.full?SSO=1 The paper from Peter et al. describes and analyses the thickness resolution of the technique. Simply, very thin parts of the cells are not detected by the method and are not contributing to the overall cell volume. Therefore, cell volume can be constant, but seen changing in the holomonitor if the cells became more rounded from a well-spread conformation. Point 3 must be shortly analyzed in their paper, especially before reaching important conclusions on the detected cell volume changes.

Reviewer 2 Report

Megha Patel et. "Evaluation of the impact of imprinted polymer 2 particles on morphology and motility of breast cancer 3 cells by using digital holographic cytometry". Manuscript was well written but findings and scientific impact still remains below the average. 

First of all, what is the basis of choosing these two cell lines? It is well-known MDA-MB-231 will be highly metastatic while MCF-7 not. Just on these two cell line experiments is not well enough to complete the conclusion. It needs further extensive multiple experiments as well as extend on a panel of cell lines. Certainly, it will depend on the author's ability. 

Line 40- Shape shifted instead of shapeshifted.

All the numerical digit need careful editing. for example flowcytometry its is 14.2% instead of 14,2% also the same in the table.

it is also better to show the live cells or real cells image rather than computer-aided. 

Figure-3- Rearrange the label legend or rearrange the picture.

Figure 4- It will be better translate absorbance into the number of cells which will help real calculation of how many cells or % of cells increased or decreased. 

for statistic analysis use multiple symbols to easy to understand the comparison like a,b,c or #

MCF-7 72h recheck statistics, since it has high SD value from the graph, is visible.

Round 2

Reviewer 2 Report

The author did not focus on the manuscript well yet regardless of scientific merit. Still, graph numerical vale remains a problem as it is figure 4. 

Author insisting to keep only holographic pictures regardless of showing live microscopic pictures.